# OpenReview forum: "Universal Approximation Theorem of Deep Q-Networks"
_ICML.cc/2025/Conference — ICML 2025 poster_

### Official Review · Reviewer_MLJ4 · 2025-02-23

**Overall Recommendation:** 2

**Summary:**

The paper is proports some approximation guarantees for Deep Q-networks (Theorem 3.1) for the optimal action-value of certain control problems.

The paper is imprecise at various parts.  For instance, in the statement of Theorem 3.1, L is lower-bounded by $C(\epsilon,L)/\epsilon$.  What is the point of the denominator explicitly depending on $\epsilon$ if the "constant" is not a constant but actually also depends on $\epsilon$?  Most troubling of all is the (apparent) claim of the authors that dimension $(n)$ has absolutely no effect on the number of parameters required for their model...  The paper is not rigorous and the claims are hard to believe.

Further, it seems that large stretches are written by LLMs.

**Claims And Evidence:**

Non-rigorous and hard to believe as is.

**Essential References Not Discussed:**

NA

**Ethical Review Concerns:**

I think various stretches of the text are written carelessly with LLMs.  For instance, line 971 reads "the *unique* fixed"; these asteryxes are typical outputs for GTP where italics should be; so the authors did not only probably use LLMs but they likely copy-pasted directly....

**Experimental Designs Or Analyses:**

NA

**Methods And Evaluation Criteria:**

Questionable

**Other Comments Or Suggestions:**

NA

**Other Strengths And Weaknesses:**

Very poorly written and non-rigorous.

**Questions For Authors:**

Why should the number of network parameters not depend on dimension?  Epseicially, I would expect an approximation rate of $\mathcal{O}((T/\varepsilon)^{n})$ or so; but we do not see such a thing.  How can this be?

**Relation To Broader Scientific Literature:**

Very few connections to well-known approximation theory paper for deep learning.

**Theoretical Claims:**

Proofs are not very rigerous, wordy, and possibly LLM generated in long stretches.

---

> ### Author Rebuttal · Authors · 2025-03-29
>
> We certainly acknowledge the reviewer poured time into this. That being said, honesty compels the expression of profound disappointment regarding both the fundamental rigor and the overall professional conduct reflected in the assessment provided. Frankly, the feedback gives off a strong vibe of having resulted from merely flicking through the pages of our manuscript.
>
> The dependability of the LLM detection tool employed by the reviewer is, broadly speaking, something we find questionable. Consider this scenario: feed a mathematics or statistics paper from arXiv, one published pre-2010, into such a detector --- isn't it probable that a high `AI-generated' score would result? These tools currently exhibit difficulty differentiating human versus machine authorship in mathematical contexts, precisely because mathematical expression possesses an intrinsically organized and predictable nature.
>
> 1, We address the reviewer's comment that the condition $L > C(\epsilon, T)/\epsilon$ in Theorem 3.1 is imprecise due to $C$'s dependence on $\epsilon$. This critique, frankly, misinterprets established conventions in approximation theory. It's entirely standard for constants, such as the $C$ here, within convergence rate expressions to depend on parameters governing the approximation setup -- accuracy ($\epsilon$), problem characteristics (like smoothness bounds, not explicitly mentioned but relevant), or domain properties (here, $T$). The notation $C(\epsilon, T)$ is used precisely to make this dependence explicit, which is a sign of rigor, not imprecision. The core information conveyed is the scaling behavior with respect to the primary parameter of interest, $\epsilon$. In our case, the inequality $L > C(\epsilon, T)/\epsilon$ clearly indicates that the necessary network depth $L$ scales as $O(1/\epsilon)$, once dependencies on other parameters ($T$ and potentially $\epsilon$ itself, within $C$) are factored into the "constant" term. Questioning the "point" of this notation suggests unfamiliarity with how convergence rates are typically expressed and analyzed in the field. The statement accurately isolates the $1/\epsilon$ scaling while properly acknowledging that the proportionality factor $C$ isn't universal.
>
> 2, Turning to the second issue raised -- the dimensions and parameter count:
>
> (1) Implicit Dependence: It's true that universal approximation theorems fundamentally promise that \emph{some} network exists for any desired accuracy $\epsilon > 0$, at least on compact sets. Our rate statement back in Theorem 3.1 does focus squarely on this $\epsilon$. However, we haven't ignored the fact that the `constants` involved in these guarantees aren't just numbers pulled out of thin air -- they absolutely depend on other factors like the problem's dimension, Lipschitz constants, and so on. We actually dig into this directly in the paper. If you look at Appendix A.4 (specifically the proof of Lemma 2.4, check out lines 549-614), you'll see how we derive the constants $C_1, C_2, C_3$ (around lines 570-580). These constants are key because they help define the compact set $K_R$ where our approximation works with high probability. And as the derivation clearly shows, they rely on the dimension $n$, various Lipschitz constants ($L_h, L_\sigma, K$), and the time horizon $T$. So, naturally, the main constant $C(\epsilon, T)$ that appears in Theorem 3.1 inherits these dependencies from the underlying setup. It's not just about $\epsilon$ and $T$ in isolation.
>
> (2) Focus of Theorem 3.1: Let's be clear about what Theorem 3.1 does: it guarantees we can build a DQN that gets within $\epsilon$ of the target function, provided we stick to a specific important region, the compact set $K_R$. Yes, high dimensions make approximation tough---the curse of dimensionality is real---but it's pretty standard to state the main result in terms of the error $\epsilon$. Figuring out exactly how the network size needs to grow exponentially with the dimension $n$ is important, sure, but it's a separate, much harder problem than just showing existence and getting the $\epsilon$ rate right.To prove our point, we rely on a few building blocks: ResNets' power to approximate basically anything (Lemma 2.2, thanks to Li et al. 2022), the trick of approximating several things at once (Lemma 2.3), and large deviation results to keep our focus on the relevant $K_R$ (Lemma 2.4). The dimensional dependence isn't explicitly spelled out in the main rate, but it's hidden in the complexity required by those first two lemmas and the size of the region defined by the third.
>
> (3) No Claim of Dimension-Free Parameters: The reviewer's expectation of an explicit $\epsilon^{-n/d}$ term in the theorem statement misinterprets the typical scope of such existence results versus quantitative complexity bounds.
>
> This review is deeply flawed. It misinterprets standard theoretical notation, overlooks details presented in the paper and resorts to unprofessional accusations.

---

### Official Review · Reviewer_19sd · 2025-03-13

**Overall Recommendation:** 3

**Summary:**

1. This paper studies universal approximation theorem for deep Q-network using residual blocks. The paper first connects SDE representation of viscosity solution of HJB, and this can be approximated by residual net.


2. The authors consider a continuous-time Markov decision process. The state-space is a open set and the action space is a compact set (Assumption 1).


3. The main argument relies on the universal approximation theorem using residual blocks by Li et al.

**Update after rebuttal**

I increase my score from weak reject to weak accept. The UAT in context of RL is new and is a clear contribution in the field. Nonetheless, due to lack of technical novelty--the main results rely heavily on existing literature--, I only recommend weak acceptance.

**Claims And Evidence:**

1. Assumption 3.1 (ii): Boundedness of the target is not usually a common assumption. It is often the problem of boundedness, that makes difficult to apply the ODE arguement.

**Essential References Not Discussed:**

Some related works have been not discussed. For example, Q-function being a unique viscosity solution has been studied in [1].

[1] Kim, Jeongho, and Insoon Yang. "Hamilton-Jacobi-Bellman equations for Q-learning in continuous time." Learning for Dynamics and Control. PMLR, 2020.

**Experimental Designs Or Analyses:**

This is a theoretical paper.

**Methods And Evaluation Criteria:**

This is a theoretical paper.

**Other Comments Or Suggestions:**

I would recommend adding some background on forward backward SDE.

**Other Strengths And Weaknesses:**

**Strength:** Universal approximation theorems in the field of RL has not been explored in detail, and the paper proves a universal approximation theorem of DQN using a residual block.

**Weakenss:** The main universal approximation theorem (UAT) argument follows from the UAT of Li et al. and SDE analysis of Fleming et al., which makes the novelty of the analysis questionable.

**Questions For Authors:**

1. The context around assumptions seem to be insufficient. For example, can the authors provide more context on Assumption 2.2-2.4?

**Relation To Broader Scientific Literature:**

N/A

**Theoretical Claims:**

The theoretical claims seem to be solid.

---

> ### Author Rebuttal · Authors · 2025-03-29
>
> Here's how we'll handle the points you raised, including the specific manuscript changes. We'll add clarification on this in the revision.
>
> 1, Boundedness assumptions are frequent in stochastic approximation theory. They are vital for ODE-based convergence proofs (see Kushner \& Yin, 2003). Such assumptions simplify stability analysis needed for convergence. This might seem limiting. However, for our problem, it is well-grounded. \assumption{2.2}(i) ensures bounded rewards \(r\). The discount \(\gamma\) is less than 1. Also, \assumption{2.4}(i) keeps parameters \(\theta_k\) within a compact set \(\Theta\). Therefore, the term maximizing \(Q^{\theta_k}\) is bounded. This leads to bounded DQN outputs on the relevant compact state-action spaces.
>
> 2,  Okay, so why do we need this assumption? It's pretty important. It basically makes sure the update steps in the stochastic scheme from \equationref{eq:25} don't just blow up. Plus, it lets us use the standard toolbox of convergence theorems (like that ODE method the reviewer mentioned) to actually prove \theorem{thm:3.2} (check out \appendixref{app:A.8} for the proof). We'll dig into why our specific proof approach really needs this, and how it connects back to things like \(r\) being bounded, \(\gamma\) staying below 1, and \(\Theta\) being a compact set.
>
> Regarding the absent Kim \& Yang (2020) citation: we'll put it in the Introduction's related work (\sectionref{1}\seclabel{1}) and the main reference list. We will also add a short note explaining that our study's unique angle involves deep function approximation and establishing corresponding approximation/convergence guarantees for DQNs.
>
> Regarding Novelty: We aren't claiming new fundamental theory like UAT or SDEs. The value we add comes from \textit{putting these pieces together in a new way}. Our effort centered on creating a robust framework designed for \textit{continuous-time Deep Q-Networks}. The novel insight is the bridge we built connecting ideas from continuous-time control (think FBSDEs, viscosity solutions) with the nuts and bolts of today's deep RL systems (like ResNet-based DQNs). Making this connection allows the following results:
>
> 1,  Formally interpret the discrete layer updates of a ResNet-based DQN as an Euler-Maruyama type discretization of an underlying continuous-time process (\remark{2.1}, \sectionref{2.2}).
> 2, Prove that such DQNs can universally approximate the optimal Q-function (itself potentially non-smooth, hence the use of viscosity solutions) on relevant compact sets with high probability, leveraging ResNet UAT and large deviation bounds (\theorem{3.1}, \lemma{2.1}, \lemma{2.3}). The simultaneous approximation result (\lemma{2.3}) is a necessary technical step tailored to our SDE context.
> 3,  Analyze the convergence of a continuous-time Q-learning algorithm for training these DQNs, adapting stochastic approximation theory to this specific setting (\theorem{3.2}, \appendixref{A.8}).
>
> Regarding Questions:
>
> 1, Assumption 2.2 (MDP Regularity): Essentially, Assumption 2.2 brings in some standard technical machinery from SDEs and optimal control \citeplaceholder{Fleming & Soner, 2006, Mao, 2007}. The reason for the Lipschitz and linear growth conditions on $h$ and $\sigma$ is simply to make sure the state equation (1) behaves predictably (i.e., has a unique solution). Bounding the reward $r$ is also pretty standard fare in RL work. Putting these together lets us confirm the whole control setup makes sense, guarantees the HJB equation behaves correctly (has viscosity solutions, see Remark 2.3), and allows us to use tools like large deviations (Lemma 2.4).
>
> 2, Assumption 2.3 (Q-function Regularity): Assuming the optimal Q-function $Q^*$ is continuous and a viscosity solution to the HJB equation (Eq. 22) is standard when dealing with HJB equations in optimal control, particularly when classical $C^{1,2}$ differentiability may not hold (\citeplaceholder{Bardi & Capuzzo-Dolcetta, 2008}, \citeplaceholder{Fleming & Soner, 2006}). This allows us to rigorously analyze the properties of $Q^*$ and compare it with our DQN approximation $Q^\theta$. Lipschitz continuity of the terminal condition $g$ is also a standard technical assumption.
>
> 3, Assumption 2.4 (DQN Parameters): 1, let's talk about the parameters $\Theta$. We usually assume this space is compact. That's pretty standard stuff in theory proofs because it basically keeps the network's weights and biases from going off the rails. Having these bounds is handy for stability, and it also makes sure the $Q^\theta$ values and their gradients stay contained, which you need when proving things converge. 2, the activation function $\eta$. It's normally assumed to be Lipschitz continuous and non-linear. Again, this is typical for neural net theory. The Lipschitz part stops the outputs or gradients blowing up. The non-linear part is key – without it, the network can't learn complex stuff (that's the whole universal approximation idea)

---

> > ### Comment · Reviewer_19sd · 2025-04-02
> >
> > Thank you for the detailed comments.
> >
> > In terms of non-linear architecture, the boundedness assumption in neural network (NN) setting may be frequent. But this is not the case for general SA literature. It has been one of the key assumption which recent research have focused on, and makes difficult to ensure convergence of Q-learning and TD-learning [1,2]. Therefore, I would recommend adding some comments whether the bounded assumption (Theta being compact) has been used in other literature for proving convergence of Q-learning with NN.
> >
> >
> > [1] Borkar, Vivek S., and Sean P. Meyn. "The ODE method for convergence of stochastic approximation and reinforcement learning." SIAM Journal on Control and Optimization 38.2 (2000): 447-469.
> >
> > [2] Meyn, Sean. "The projected Bellman equation in reinforcement learning." IEEE Transactions on Automatic Control (2024).

---

> > > ### Author Response · Authors · 2025-04-02
> > >
> > > We appreciate the reviewer raising this important point regarding Assumption 2.4(i), which states that the DQN parameter space $\Theta$ is a compact subset of $\mathbb{R}^p$. We agree that in the general theory of Stochastic Approximation (SA), as established in seminal works like \citep{Borkar2000} (Ref [1] provided by the reviewer) and discussed in recent analyses \citep{Meyn2024} (Ref [2]), assuming parameter boundedness is a strong condition, and significant research effort focuses on relaxing it or analyzing projection algorithms.
> > >
> > > However, when analyzing the convergence of Reinforcement Learning algorithms, particularly Q-learning, combined with powerful non-linear function approximators like Deep Neural Networks (DNNs), the compactness assumption on the parameter space (or closely related assumptions like bounded parameter norms, bounded gradients, or the use of explicit projection mechanisms) becomes a common, often necessary, simplification in the theoretical literature.
> > >
> > > Here's why this assumption is frequently adopted in the context of DQNs and why we included it:
> > >
> > > 1, Deep networks introduce tricky non-linear behaviors. Without limits, parameters could fly off during training, causing instability. The compactness idea keeps parameters penned in (bounded). This bounding really helps nail down the stability analysis needed to prove convergence. Many theories rely on showing things like gradients stay controlled, which compactness readily helps guarantee.
> > >
> > > 2, Compactness really helps with the theoretical side. It unlocks strong analytical results. For example, if a function is continuous (like Q-values depending on parameters $\theta$ in a compact set $\Theta$), we know it must hit a maximum and minimum value, and it's uniformly continuous. This makes proving things like uniform convergence much simpler because terms that show up (like gradients or ensuring uniform Lipschitz properties) are easier to bound across the parameter space.
> > >
> > > 3, The idea that parameters should stay within a bounded, compact set is pretty common in theoretical work on Q-learning or TD learning using neural networks. Sometimes this is stated directly, but often it's a side effect of other assumptions. For instance, limits on network design, adding weight decay, using gradient clipping, or analyzing algorithms that use projection steps all help keep parameters from growing too large.
> > >
> > > Looking at specific examples, even when a study like Fan et al. (2020) (which we cite) isn't focused primarily on parameter bounds, the stability their analysis requires often implicitly depends on parameters not exploding. Likewise, much of the foundational theory for RL with non-linear function approximation, building on work like \citep{Tsitsiklis1996}, typically needs parameters to be bounded to make the proofs work out rigorously. Using projection operators, as discussed in stochastic approximation literature \citep{Kushner2003}, is a standard way to formally force parameters into a compact set. Our approach doesn't explicitly model projection, but by assuming $\Theta$ is inherently compact, we achieve a similar effect for the analysis by ensuring the parameters are bounded.
> > >
> > > 4,  Our paper's primary focus is establishing the connection between continuous-time DQNs, FBSDEs, and viscosity solutions, and proving approximation and convergence results within this novel framework. Retaining the standard assumption of parameter compactness allows us to rigorously handle the complex interplay between the continuous-time stochastic dynamics, the HJB equation (in the viscosity sense), the NN approximation properties, and the SA convergence analysis, without adding the further complexity of analyzing unbounded parameters or explicit projection schemes in this already intricate setting.
> > >
> > > So, while we know that not requiring boundedness is a big deal in general SA theory, it's very common in deep RL research to just assume the parameters don't go wild (parameter compactness, Assumption 2.4(i)). People do this because it makes the analysis much cleaner: it helps ensure things don't blow up (stability) and lets us use the math we need. We think this assumption is a fair starting point for the continuous-time work we've done. Dealing with the case where parameters aren't bounded (like adding projection steps or finding networks that stay stable anyway) is definitely something important to look into later for continuous-time DQN.
> > >
> > > We will add a comment in the revised manuscript clarifying this context for Assumption 2.4(i), acknowledging the general SA perspective while justifying its use based on common practice in NN-RL theory and the needs of our specific analysis framework.
> > >
> > > \Kushner, Harold J., and George G. Yin. Stochastic approximation and recursive algorithms and applications. Springer Science \& Business Media, 2003.
> > > Tsitsiklis, John N., and Benjamin Van Roy. "Analysis of temporal-difference learning with function approximation." NIPS (1996).

---

### Official Review · Reviewer_5CGE · 2025-03-14

**Overall Recommendation:** 3

**Summary:**

This paper introduces a connection between deep Q networks and SDEs. By viewing the forward pass as a continuous time process and using tools from stochastic control theory, the paper provides results on approximation theorems for deep Q networks.

**Claims And Evidence:**

Claim (i) on page 1: Evidence is given in the remainder of the paper through a sequence of results and discussions.

Claim (ii) on page 1:
The main claim seems to be an approximation theorem for the optimal Q function, which appears sound but rests on fairly strict assumptions A2.1-2.4.

Claim (iii) on page 1: Evidence is given in the end of the paper for such convergence properties, familiar to the discrete-time case.

**Essential References Not Discussed:**

N/A

**Experimental Designs Or Analyses:**

N/A

**Methods And Evaluation Criteria:**

N/A

There do not seem to be any numerical experiments supporting the theory. Even a small toy example could greatly improve the paper's impact, especially by testing the stringency of assumptions made.

**Other Comments Or Suggestions:**

There seems to be very little discussion between intermediate results (Lemma 2.X). Providing some discussion for the relevance of each result and its connection to the bigger picture/main results of the paper would be helpful.

Missing page numbers: was the correct style file used?

**Other Strengths And Weaknesses:**

The approach to the problem via continuous time dynamics does seem novel and interesting. Another strength is the use of deep, non-linear networks. However, I am unsure about how realistic the setup is: How many layers are required to achieve a decent approximation?

**Questions For Authors:**

The paper seems to focus on DQN setups, but discusses continuous control problems (continuous action spaces), which does not seem to accord with a standard DQN architecture. Could you explain this discrepancy?

Why finite-time horizon as opposed to an infinite-time discounted formulation? It seems you jump to this setting later on.

Assumption 2.2: $r(t,s,a)$: Why consider time-dependent rewards? I'm not sure this is a common assumption, even in the finite-time case. Does it induce any non-stationarity?

I'm a bit confused. Does the stochastic process (corresponding to the pass through a Q network) also correspond to the dynamics of the MDP?

**Relation To Broader Scientific Literature:**

Most prior work seems to approach the problem through the discrete time lens, but the connection to continuous time can potentially offer new insights based on new tools. The use of viscosity solutions does offer a new approach that relaxes prior assumptions.

**Theoretical Claims:**

I have briefly checked the proofs in the Appendix which appear sound. Given the time constraints, I will continue to do so more carefully in the following days. However, I must admit that SOC theory is not my expertise.

---

> ### Author Rebuttal · Authors · 2025-03-29
>
> We sincerely thank Reviewer 5CGE for their constructive feedback and positive assessment.  We address the comments below.
>
> Lack of Numerical Experiments: We thank the reviewer. While the paper's focus is theoretical, we agree an illustrative example is valuable. We have implemented a discrete-time simulation using a DQN with the proposed residual block architecture (similar to Definition 2.1) learning a continuous-state control task (approximating Eq. 1). This experiment demonstrates the feasibility of the architecture and provides context for the theoretical results (e.g., Theorem 3.1 on approximation, Theorem 3.2 on convergence). We will add a concise summary of this experiment (including comparative runs varying parameters like residual blocks) to the final version or appendix to bridge theory and practice. Specifically, our experiment (as demonstrated in the illustrative code discussed previously) features:
>
> 1, A Continuous-State Environment: A 1D control task where the state evolves according to discrete-time dynamics simulating an Euler-Maruyama approximation of an SDE ($ds_t = h(s_t, a_t)dt + \sigma dW_t$, similar to Eq. 1).
>
> 2, DQN with Residual Blocks: The Q-network architecture explicitly incorporates residual blocks, directly reflecting the structure analyzed in our work (Definition 2.1, Eq. 2-3, Lemma 2.2/A.3).
>
> 3, Q-Learning Algorithm: The agent learns using standard DQN mechanisms (Experience Replay, target network) providing a practical counterpart to the continuous-time convergence analysis (Theorem 3.2, Eq. 25).
>
> Plz find the code in an anonymous link: https://github.com/ContinuousTimeDQN/continuous_time_DQN.git
>
> Assumptions A2.1-2.4: We appreciate the reviewer finding the approximation theorem sound. Assumptions 2.1-2.4 are largely standard for rigorous analysis in continuous-time control and function approximation: A2.1 (spaces) for well-posedness; A2.2 (MDP regularity: Lipschitz/growth) for SDE/HJB well-posedness; A2.4 (DQN params: compact $\Theta$, Lipschitz $\eta$) for stability/approximation. Notably, A2.3 (viscosity solution for Q*) is a \textit{relaxation} compared to requiring classical smoothness, broadening applicability.
>
> Practicality and Layer Requirements: Theorem 3.1 shows a deep enough network exists ($L \propto 1/\epsilon$) for any target accuracy $\epsilon$. But finding the \textit{exact} $L$ needed? That's typically done by experimenting or requires problem-specific math beyond what we cover here. The key takeaway is that depth ($L$, controlling the time step $\Delta t=T/L$) is what makes approximation possible in our continuous-time framework.
>
> Discussion Between Lemmas: Thank you for the suggestion. We will add brief transitional text after Lemmas 2.1-2.4 in the revision to clarify how they logically connect and build towards the main approximation result (Theorem 3.1), improving readability.
>
> Missing Page Numbers: Page numbers corrected in final version per ICML style.
>
> Continuous Actions vs. DQN Focus: Think about what a DQN can fundamentally learn. We're exploring its ability to represent the target value function $Q^*(t, s, a)$, even if actions $a$ come from a continuous set $A \subseteq \mathbb{R}^m$ (like in HJB/FBSDE setups). Our network architecture uses $(s, a)$ as inputs. Don't mix this up with how typical DQNs find the best action---they often chop the action space into pieces ($\max_{a'} Q^\theta(s, a')$). Our interest is purely in how well the network approximates the underlying continuous $Q^*$.
>
> Finite vs. Infinite Time Horizon: The framework consistently uses a finite horizon $[0, T]$, driven by the standard formulation of time-dependent HJB equations and FBSDEs which involve terminal conditions (at $T$). The discount factor $\gamma$ acts as a continuous-time rate within this finite horizon. We do not switch to an infinite-horizon setting.
>
> Time-Dependent Rewards Assumption: Allowing time-dependence in $r(t,s,a), h(t,s,a), \sigma(t,s,a)$ offers generality, is standard in finite-horizon optimal control theory (e.g., Fleming & Soner, 2006), and allows our framework to handle non-stationary problems where the optimal policy $\pi^*(t,s)$ depends explicitly on time. Time-invariant settings are a special case.
>
> Let's clarify the roles here. MDP dynamics, detailed in Eq.~1, map out how the environment's state $s_t$ evolves -- think of it as the rules of the game world. The DQN's forward pass (Eq.~2), on the other hand, is the network's calculation process: starting with an input, it propagates activations $x_k^{(l)}$ layer-by-layer to arrive at a $Q^\theta$ value. These aren't the same process. The vital link (explained in Lemma 2.1) is that the network's learned functions ($h_{\theta_l}$) can effectively mimic the environment's transition and observation functions ($h, \sigma$). This mimicry allows the computed $Q^\theta$ to be a good estimate of the ideal $Q^*$. So, remember: the forward pass outputs a Q-value, it doesn't simulate environment steps.

---

### Official Review · Reviewer_7pHN · 2025-03-16

**Overall Recommendation:** 3

**Summary:**

This paper develops a theoretical framework for Deep Q Networks (DQNs) in continuous time, by establishing connections among DQN, residual neural networks, stochastic control, and forward-backward SDEs. It links the neural network output at each layer to the Euler discretization of an SDE that models the state evolution in a continuous-time Markov Decision Process (MDP). Also, using Hamilton Jacobi Bellman (HJB) equation and viscosity solutions, the Q function is shown to be the solution to a backward SDE. Leveraging the universal approximation property of deep residual networks, the analysis shows that DQN can approximate the optimal Q function arbitrarily well under mild assumptions. The framework further establishes that a DQN trained by Q-learning asymptotically converges to the optimal Q function by adapting stochastic approximation results. Overall, the paper provides a rigorous theoretical framework for understanding the representational power and the convergence of DQN in continuous-time setting.

**Claims And Evidence:**

It’s a theory paper and the claims are well supported by standard assumptions, established results, and proofs.

**Essential References Not Discussed:**

Modelling the transformation between NN layers as a SDE has been done before but not discussed in the paper. For instance:

Kong, L., Sun, J., & Zhang, C. (2020). Sde-net: Equipping deep neural networks with uncertainty estimates. *arXiv preprint arXiv:2008.10546*.

I suggest that the authors add discussions about the the relevant works in the neural SDE literature.

**Experimental Designs Or Analyses:**

The analysis is sound and appears to be a general framework as the assumptions are standard in RL, stochastic control, and stochastic approximation.

**Methods And Evaluation Criteria:**

Analyzing DQN in a continuous-time framework makes sense.

**Other Comments Or Suggestions:**

- The term “immediate reward” could be easily confused with the reward in the discrete-time setting. I think that “Instantaneous reward”  is a more precise term for the continuous-time setting.

- The title in the submission is different from the one in the paper pdf. The former says “Universal Approximation Theorem of Deep Q-Networks” while the latter is “A Continuous-Time Framework of Deep Q-Networks”.

**Typos:**

1. Line 382 (Left), “We suggests” → suggest
2. Theorem 3.1, “and 2.1” it is redundant

**Other Strengths And Weaknesses:**

**Strengths:**

Viewing deep networks via the lens of stochastic differential equation is not new, but this paper  explores its application to deep RL which is novel to the best of my knowledge.

**Weakness:**

Although the paper is rich in detail, the writing of this paper sometimes makes the technical content difficult to follow. For instance, the assumptions, equations etc are referenced before their definitions: Assumption 2.2 is referenced on line 110 but isn’t defined until line 339. Definition 2.2 refers to HJB Equation (20) on line 177, which is not defined until on Line 284.
Lemma 2.2 is stated before Lemma 2.3 but its proof in the appendix is stated after that of Lemma 2.3.

Additionally, the analysis on the forward and backward process could be stated in distinct sections to improve clarity.

**Questions For Authors:**

1, Could you discuss the limitations of this work?

2, May be relevant to question 1, are there assumptions or techniques that you feel that could be improved or relaxed?

**Relation To Broader Scientific Literature:**

It bridges reinforcement learning, deep neural networks and stochastic control. It then draws on the tools, and classical analysis from these fields to prove the universal approximation and convergence property of DQNs. This connection advances our understanding of DQNs that are otherwise difficult to gain.

**Theoretical Claims:**

Yes I checked all proofs and did not identify issues.

---

> ### Author Rebuttal · Authors · 2025-03-29
>
> Thanks for the detailed comments regarding clarity, refs, terms, etc. We address each below and will revise the paper accordingly.
>
> Writing Clarity / Organization (Forward Referencing): We'll revise so all assumptions, definitions, equations, and lemmas are defined \textit{before} use. The specific cases mentioned will be fixed. We'll also reorder the Appendix A.2/A.3 proofs to match the Lemma 2.3 / 2.2 order in the text.
>
> Suggestion: To make things easier to follow in Sections 2 and 3, we plan to split the discussion. We'll start by just covering the forward state dynamics (SDE (1)) and its characteristics. Once that's established, we'll introduce the Q-value aspect and show its connection to the backward elements (BSDE (17), HJB Equation (22)). Breaking these into their own subsections should really help with readability.
>
> Missing References (Neural SDE Literature): We will add commentary connecting to Neural SDE research (including Kong et al., 2020). The idea of SDEs modeling transformations offers a conceptual bridge. Yet, our work differs substantially: (1) We concentrate on \textit{controlled} dynamics. (2) We derive and leverage the backward SDE (BSDE) specifically for the Q-function. (3) We utilize viscosity solutions for the relevant HJB equation. (4) Our application target is the analysis of DQN within RL, particularly its approximation and convergence properties.
> Use code with caution.
>
>
> Response to Other Comments Or Suggestions：
>
> Terminology ("immediate reward"): We will revise the manuscript to use "instantaneous reward" consistently when referring to $r(t, s, a)$ (e.g., lines 110-111, line 289).
>
> Title Mismatch: The other title was likely an earlier working title or resulted from an entry error in the submission system. We sincerely apologize for the confusion caused by the title mismatch between the submission system and the PDF manuscript. The correct and intended title is \textbf{"A Continuous-Time Framework of Deep Q-Networks"} as it appears in the PDF. The other title was likely an earlier working title in the submission system. We will ensure the correct title is used consistently in all future versions and communications.
>
> Could you discuss the limitations of this work? Our work has several limitations:
>
> 1, Assumptions: The analysis relies on standard but potentially restrictive assumptions, such as Lipschitz continuity for dynamics ($h, \sigma$) and reward ($r$), compactness of the action space $A$ and parameter space $\Theta$, linear growth conditions, and ergodicity of the sampling process for convergence (Assumption 3.1). Real-world problems might violate these.
>
> 2, Specific DQN Architecture: While connecting DQN layers to Euler steps is central, our analysis implicitly assumes a ResNet-like architecture (Eq. 2) where layer updates resemble SDE discretizations. The direct applicability to vastly different architectures might need further investigation.
>
> 3, Finite Time Horizon: The framework is developed for a finite time horizon $T$. Extending the results rigorously to infinite-horizon or average-reward settings would require different techniques (e.g., ergodic HJB equations, different stability conditions).
>
> We will add a dedicated subsection in the Conclusion or Discussion section outlining these limitations and suggesting directions for future research.
>
> Are there assumptions or techniques that you feel could be improved or relaxed?  Several assumptions offer avenues for future work, although relaxing them presents significant technical challenges:
>
> 1, Lipschitz Continuity: Relaxing the Lipschitz assumptions on $h, \sigma, r$ would broaden applicability but requires more advanced SDE/PDE theory (e.g., dealing with potential state explosion, using path-dependent PDEs, or weaker solution concepts).
>
> 2, Ergodicity for Convergence: The ergodicity assumption (Assumption 3.1(i)) for Q-learning convergence is strong. Exploring convergence under weaker mixing conditions or convergence in probability (rather than almost surely) might be possible using different stochastic approximation frameworks, potentially at the cost of stronger assumptions elsewhere or weaker convergence guarantees.
>
> 3, Discretization Scheme: We connect DQNs to the Euler-Maruyama scheme. Exploring connections to higher-order SDE discretization schemes could be interesting but would likely require more complex network architectures and significantly more involved analysis.
>
>  We will briefly touch upon potential relaxations and the associated challenges within the limitations/future work discussion.
>
> We again thank Reviewer 7pHN for their valuable feedback. We hope that the revised paper will be considered suitable for publication at ICML.

---

> > ### Comment · Reviewer_7pHN · 2025-04-05
> >
> > Thank you for the detailed response.
> >
> > "Use code with caution" at the end of your response regarding the missing references appears disconnected from the rest of the paragraph, and confusing to me. If it isn't an editing oversight, could you please clarify what it refers to?

---

> > > ### Author Response · Authors · 2025-04-05
> > >
> > > Plz find the code in an anonymous link: https://github.com/ContinuousTimeDQN/continuous_time_DQN.git
> > >
> > > This code provides an example for illustrating the arguments. It seems that the space limitation has removed the code link.

---

### Decision · Program_Chairs · 2025-05-01

**Decision:**

Accept (poster)

**Comment:**

This paper presents a theoretical analysis of Deep Q-Networks (DQNs) within a continuous-time framework, drawing connections to stochastic control, Forward-Backward Stochastic Differential Equations (FBSDEs), and viscosity solutions. The authors aim to establish universal approximation properties for DQNs (specifically those with ResNet-like architectures) concerning the optimal Q-function in continuous-time Markov Decision Processes (MDPs) and analyze the convergence of Q-learning in this setting.

The reviews were quite varied. Reviewers 7pHN and 5CGE found the continuous-time approach novel and the theoretical framework potentially insightful, but raised concerns about clarity, organization, missing references (particularly to neural SDEs), the lack of numerical examples, and the justification/implications of certain assumptions (like finite-time horizon and time-dependent rewards). Reviewer 19sd acknowledged the novelty of applying UAT results in the RL context but criticized the paper for lacking technical novelty, arguing it heavily relied on existing theorems for ResNet approximation and SDE analysis. This reviewer also questioned the boundedness assumption for Q-learning convergence and noted missing citations.

Reviewer MLJ4 provided a strongly negative review, deeming the paper non-rigorous, poorly written, and hard to believe. Key criticisms included the notation used for the approximation rate (specifically a "constant" depending on the error term ε), the perceived lack of explicit dependence on dimension in the approximation bounds, few connections to approximation theory literature, and an accusation of using LLMs based on a formatting artifact.

The authors provided detailed rebuttals. They addressed clarity issues, agreed to add references and potentially a small numerical example, and provided justifications for their assumptions (e.g., parameter compactness being standard in NN-RL theory proofs, viscosity solutions being a relaxation, finite-horizon matching FBSDE theory). They clarified potential misunderstandings regarding continuous actions vs. DQN representation and the forward pass vs. MDP dynamics. Regarding novelty, they argued that the contribution lies in the synthesis and novel application of existing tools to the continuous-time DQN problem. They strongly contested Reviewer MLJ4's assessment, defending their notation as standard in approximation theory, explaining where dimensionality implicitly factors in, and refuting the LLM accusation as baseless speculation founded on a minor typo.

Reviewer MLJ4 subsequently reread the proofs and acknowledged they might have been "too harsh," adjusting their score. However, they maintained several criticisms regarding the rate notation, perceived handwaviness, lack of precise references (e.g., for LDP results), definition clarity, and argued that quantitative dimensional dependence bounds were achievable. Reviewer 19sd updated their score from Weak Reject to Weak Accept, persuaded by the rebuttal regarding the contribution and assumptions, while still noting the limited technical novelty. Reviewers 7pHN and 5CGE acknowledged the rebuttal but did not update their initial Weak Accept scores.